# Warming impacts potential germination of non-native plants on the Antarctic Peninsula

Stef Bokhorst [1✉], Peter Convey [2], Angélica Casanova-Katny [3] & Rien Aerts[1]

The Antarctic Peninsula is under pressure from non-native plants and this risk is expected to increase under climate warming. Establishment and subsequent range expansion of non-native plants depend in part on germination ability under Antarctic conditions, but quantifying these processes has yet to receive detailed study. Viability testing and plant growth responses under simulated Antarctic soil surface conditions over an annual cycle show that 16 non-native species, including grasses, herbs, rushes and a succulent, germinated and continued development under a warming scenario. Thermal germination requirement (degree day sum) was calculated for each species and field soil-temperature recordings indicate that this is satisfied as far south as 72° S. Here, we show that the establishment potential of non-native species, in number and geographical range, is considerably greater than currently suggested by species distribution modelling approaches, with important implications for risk assessments of non-native species along the Antarctic Peninsula.

[1] Department of Ecological Science, Vrije Universiteit Amsterdam, Amsterdam, The Netherlands. [2] British Antarctic Survey, Natural Environment Research Council, Cambridge, UK. [3] Laboratorio de Ecofisiologia Vegetal y Núcleo de Estudios Ambientales (NEA), Facultad de Recursos Naturales, Universidad Católica de Temuco, Temuco, Chile. ✉email: s.f.bokhorst@vu.nl

Climate warming is generating opportunities for non-native species to be introduced via human assistance into colder biomes[1], with some of these species potentially becoming invasive. Such species may increase the local species pool but they also have unknown though generally negative consequences for ecosystem processes and native biodiversity[2]. Due to long-term isolation and the continent's extreme environmental conditions, contemporary Antarctic terrestrial ecosystems have low diversity and simplified food webs compared to areas with milder climates[3]. The Antarctic Peninsula region is considered at high risk of establishment of non-native species due to rapid climate warming[4,5] and the high number of visitors (tourist and science operations) that provide introduction vectors[6–16]. Invasions by vascular plants, of which there are currently only two Antarctic native species[17], are predicted to result in large impacts on species interactions and ecosystem process rates[18–24]. Extensive modelling exercises have concluded that up to four cold-tolerant plant species may be suitably pre-adapted to survive in the Antarctic Peninsula climate[16]. However, these approaches are based on modelled climate variables across large regions whereas the microclimatic conditions that plants experience on the ground are often poorly reflected even by nearby standard weather stations[25,26]. Furthermore, very few seed germination studies of non-native plant species have been conducted in Antarctic soils under relevant microclimatic conditions[27,28]. Therefore, testing directly whether different non-native plant species/types can germinate and survive in Antarctic soils under realistic multi-season edaphic temperature regimes will provide important new information to enhance and improve modelling approaches applied in attempts to predict establishment risk in Antarctica[12,16,29].

In this work, we aimed to quantify and compare time to germination and subsequent growth of different plant types (26 species; Supplementary Table 1) in natural soil obtained from Antarctica over an experimentally simulated full summer–winter–summer cycle. For this work, we selected species based on (1) a native distribution in the proximity of Antarctica[30], (2) ruderal characteristics[16,23], or (3) those that are already known to be invading sub-Antarctic islands[10,31]. Time to germination is an important first step as it determines the time subsequently remaining for growth before the onset of winter in these cold environments with short growing seasons. Antarctic temperature conditions were experimentally simulated in climate chambers in the Netherlands based on (a) high spatial and temporal resolution seasonal soil surface temperature records from the Antarctic Peninsula[32], and (b) a climate warming scenario where summer temperatures are raised by 5 °C, to assess the influence of warming on germination and subsequent growth (Supplementary Fig. 1). We hypothesised (1) that grasses will perform better than other plant types, based on the results of species distribution modelling approaches applied to date[10,16,29] and the current establishment of non-native grasses in Antarctica[33] and (2) that warmer temperatures result in more rapid seed germination and greater growth of all species. Through our experimental approach, we aim to improve understanding of which widespread global invaders are most likely to establish, given the opportunity, in Antarctic Peninsula terrestrial ecosystems under current and realistic future climate warming scenarios. Further, we use our results to quantify species-specific degree day sum requirements for seed germination in Antarctic soil to map their potential distribution under contemporary conditions based on available soil surface temperature recordings from the field. In addition, as future climate warming will vary across the Antarctic Peninsula region[4], we calculate soil surface degree day sums along the Antarctic Peninsula under both +3 and +5 °C warming scenarios. We find that under current climate conditions the soil surface temperature regimes along the Antarctic Peninsula are suitable for 16 non-native species from 7 different plant families. These findings have important implications for risk assessments of non-native species along the Antarctic Peninsula.

## Results and discussion

**Seed germination.** Of the 26 study species, 18 germinated in Antarctic soil, with the tree *Pinus sylvestris* only germinating under the warming scenario conditions (Supplementary Table 2). In separate trials of germination in commercial potting soil, the majority of trees and shrubs germinated at 15 °C, supporting seed viability. However, *Betula nana*, *Blechnum penna-marina* and *Larix sibirica* failed to germinate in potting soil, suggesting that for these species the lack of germination in Antarctic soil may have been due to inherently poor germination. None of the germinating species set seed during the course of the experiment under either scenario. This could suggest that these species are unable to set seed under Antarctic climatic conditions, although some would not normally reach sexual maturity on the timescale of this experiment in parts of their natural distributions. Such a scenario would not, however, prevent species persisting and expanding their populations as clonal growth is often common in colder habitats[34] and has been observed for both native Antarctic vascular plants[35–37], the spread of the non-native grass *P. annua* at various sites in Antarctica[38,39], and its congener *P. pratensis* at its single (now eradicated) occurrence site on the Antarctic Peninsula[40]. Alternatively, our finding could be a result of plants not having reached a sufficient size to set seed. Indeed, both initial germination and survival and development to larger plant size is a crucial factor for the establishment of non-native vascular plants in Antarctica. This is evident from the fact that the few seed germination experiments that were conducted in Antarctica (before the current general ban on the import of non-native species under the Protocol on Environmental Protection to the Antarctic Treaty[41]) did not show success, while transplants typically fared better[42].

Sixteen of the 18 species showed regrowth in the second summer season following exposure to simulated winter conditions with grasses growing from overwintering roots. *Cerastium arvense* had above-ground plant parts that survived the winter, which makes it a potentially high-risk invader for Antarctic terrestrial ecosystems if seeds reach the Antarctic Peninsula. For all other species/plant types (herbs, rushes and the succulent *Sedum album*) that showed regrowth, we were unable to confirm if they emerged from overwintering roots or from seeds that had not germinated in the first simulated summer season.

In the first simulated growing season, the time to germination was on average 28 days longer under contemporary compared to warmed conditions across all plant species and types (Fig. 1a, Table 1). The same pattern was observed following the simulated winter, with above-ground growth emerging 10 days earlier under warmed conditions (Fig. 1b). *Pinus sylvestris* took the longest to germinate and required the largest degree day sums (Tukey HSD $p < 0.05$), while there were no significant differences between the other plant species for duration and degree day sums (Table 1, Fig. 1). After the simulated winter, no consistent differences in time to above-ground emergence were apparent between plant species. The temperature requirements (degree day sums) for germination were generally the same for plants grown under current temperatures and the warming scenario, confirming the robustness of this approach (Fig. 1c). However, the herbs *Draba polytricha*, *Jasione montana*, *Plantago lanceolata* and *Astragalus cruckshanksii* required lower (all Tukey HSD $p < 0.05$) degree day sums under the warming scenario (Supplementary Table 2).

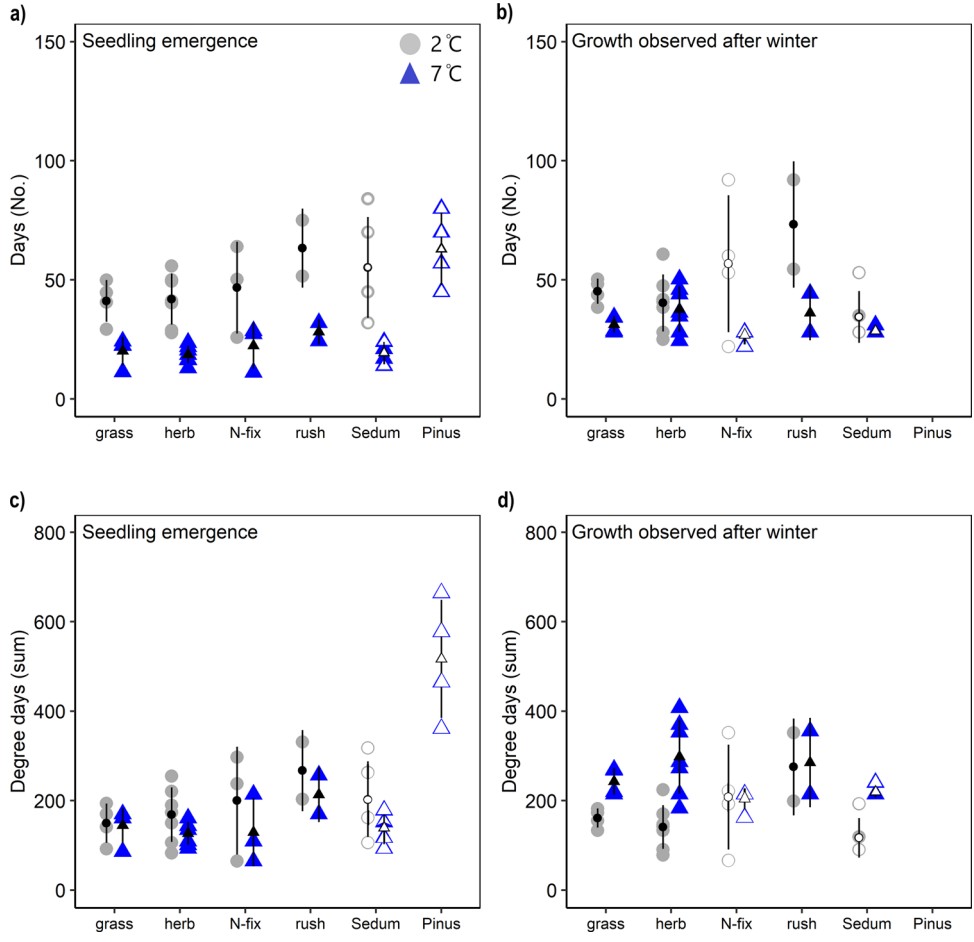

**Fig. 1 Time and temperature requirements.** Time and degree day sums required for germination of the main plant types under Antarctic growing season temperatures (2 °C; grey circles) and climate warming (7 °C; blue triangles) in Antarctic soil. **a** Number of days required to first germination and **b** number of days required for growth after winter. **c** Degree day sums required for the first germination and **d** number of degree day sums required for growth after winter. Grey circles and blue triangles represent species-specific means within each plant type (grass: 4, herb: 7, N-fixing plants: 3 and rushes: 2), with the exception of *Sedum album* and *Pinus sylvestris* where 'open' data points represent experimental pots (*n* = 4–5) and for N-fixing plants after winter when only *Astragalus cruckshanksii* grew (*n* = 4–5). Black symbols represent the mean with SD as error bars. For a complete species list see Supplementary Tables 1 and 2.

**Table 1 ANOVA-statistics (*F* and *p* values) of germination time, degree day sum requirements, plant height, and a number of plant shoots/leaves, grown in Antarctic soil at 2 and 7 °C (T) between different plant types (PT); grass (*n* = 4), herb (*n* = 7), nitrogen-fixing plants (*n* = 3), rushes (*n* = 2), succulent (*n* = 5) and *Pinus* (*n* = 5), during the first and second simulated growing season.**

| | Growing season | Temperature | | Plant type | | T × PT | |
|---|---|---|---|---|---|---|---|
| | | (1,24) | *p* | (5,24) | *p* | (4,24) | *p* |
| Number of days till germination | First | 38.1 | <0.001 | 4.5 | 0.005 | 0.5 | 0.706 |
| | Second | 6.2 | 0.021 | 1.4 | 0.277 | 2.9 | 0.043 |
| Degree day sums till germination | First | 0.4 | 0.545 | 13.0 | <0.001 | 0.3 | 0.875 |
| | Second | 60.3 | <0.001 | 4.1 | 0.012 | 2.3 | 0.087 |
| Plant height | First | 0.1 | 0.801 | 4.3 | 0.006 | 0.3 | 0.878 |
| | Second | 0.1 | 0.910 | 5.3 | 0.004 | 0.7 | 0.569 |
| Number of shoots/leaves | First | 14.0 | 0.001 | 9.8 | <0.001 | 0.1 | 0.972 |
| | Second | 4.7 | 0.042 | 9.8 | <0.001 | 0.2 | 0.936 |

See Supplementary Table 1 for the full species list and Supplementary Tables 2–4 for species-specific responses. Degrees of freedom are presented in brackets for each main factor and the interaction term. The significant Temperature × Plant type (T × PT) interaction was due to a difference (Tukey HSD *p* < 0.05) between rushes (at 2 °C) and grasses (at 7 °C).

During the second simulated growing season the average degree day sum required for plants to emerge above-ground was 73% greater for all plants growing under the warmed ($281 \pm 18$) compared to the contemporary ($162 \pm 17$) scenario, although this difference was not significant for all species (Fig. 1d, Supplementary Table 2). The different degree day sum requirements for plant emergence between the temperature treatments during the second simulated growing season reflects the short time difference (10 days) until plant emergence and may have resulted from seed preconditioning during the simulated winter[43] and/or different thermal requirements for regrowth from overwintering plant parts. The degree-day sum requirement for the *Sedum album* was lower (Tukey HSD $p < 0.05$) than for rushes after winter.

Considering that many of the tested species germinated and grew under the contemporary scenario it is likely that they will be able to do so under current field temperature conditions. The tested species represent only a small fraction of the potential species pool that could reach Antarctica through human transport[12,13,44], and the success or failure of some plant types/species does not necessarily, therefore, represent the future success of other species within certain plant functional types. Diurnal temperature fluctuations can be an additional important factor behind seed germination[43,45] indicating that various temperature modulations may need to be tested before specific species can be ruled out. However, even with these constraints, our data show that certain plant types (trees and shrubs) are less likely to establish due to a long time for plant emergence under current Antarctic conditions. This reflects in part, the temperature constraints on trees in cold regions[46].

**Plant growth**. Plants were on average 139% taller when grown under the warming scenario in the first season (Fig. 2a), which is in line with the more rapid germination (Fig. 1a) allowing for a longer post-germination growth period. In the second summer season, plants were on average double the height under the warming scenario (Fig. 3b), although this difference was not significant for all species (Supplementary Table 3). Overall, the plant growth responses were consistent with studies of plant responses to experimental warming[47] and phenological trends observed across various regions in the Northern Hemisphere[48]. Larger growth may be expected to benefit plant survival of harsh environmental conditions but, if plants become too tall, they may conversely be exposed to greater wind abrasion and have lower protection by snow cover[49,50] than do shorter stature plants.

The number of shoots/leaves did not differ under the two scenarios for the different plant types during the first growing season (Fig. 2c), indicating that other environmental factors, such as nutrient or water availability, may limit or control shoot production[51]. A number of species grew more shoots/leaves under the warming scenario, including *Trifolium repens* (49%), *Plantago lanceolata* (150%) and *Luzula spicata* (209 %). In contrast, the number of *Caiophora coronata* shoots was nearly twice as high under the contemporary scenario (Supplementary Table 4). Following the simulated winter, there was again no overall effect of temperature on the mean number of shoots/leaves for the main plant types. The grass *Deschampsia cespitosa* was the only exception, producing five times more leaves under the warming scenario (Fig. 2d, Supplementary Table 4).

**Mapping of degree day sums and potential germination**. Based simply on the degree day sum requirements for germination (Supplementary Table 2), grasses, herbs (including N-fixing plants), rushes and *Sedum album* would currently be able to germinate as far south as 72°S during the natural growing season,

but with very limited time subsequently available (0–16 days) for growth before the onset of winter (Fig. 3). Therefore, only for fast-germinating species, such as the grass *Holcus lanatus* and the herb *Taraxacum officinale*, would conditions this far south be suitable. The post-germination period available for growth (as defined by the photoperiod and temperature) was longer (60–81 days) at Anchorage Island (67°S) and was maximum (104 days) at Deception Island (62°S). Sixteen of the 26 investigated species are likely to be able to germinate and develop as far south as c. 67°S along the Antarctic Peninsula under current climate conditions (Fig. 4), adding 7 families (Brassicaceae, Campanulaceae, Crassulaceae, Juncaceae, Loasaceae, Plantaginaceae and Scrophulariaceae) to the list of potentially invasive plants. This is four times greater than species distribution modelling approaches have to date indicated[16], although some of these families were considered a risk for the Antarctic Peninsula during a recent horizon scanning exercise[10].

The declining trend of soil surface degree day sums along the Scotia Arc archipelagoes and Antarctic Peninsula (Fig. 4) is consistent with colder air temperatures as latitude increases[52]. Despite this, the extrapolation from the limited number of measuring locations available across the study region can at best provide only an estimate of the actual field conditions experienced. The measured degree day sum at Signy Island (252) and short effective growth period (59 days), despite being located at only 60°S (Figs. 3 and 4), emphasises how local microclimate conditions may be affected by factors such as cloudiness, and snow cover thickness and duration[53]. At the other extreme of the latitudinal gradient considered here, Ablation Valley (Alexander Island; 70°49′S, 68°25′W, ASPA no. 147), supports regionally rich bryophyte communities, unusual for its latitude and cold climate[54], and may provide suitable microhabitats for non-native species. Despite these limitations, the degree-day map (Fig. 4) visualises realistic soil surface degree day sums which are currently lacking in the scientific literature and forms a starting point that can only be further refined through extensive field measurements similar to work done in Taylor Valley (Victoria Land; 77°37′S 163°00′E)[55], with such data currently being unavailable in our study region.

The soil surface degree day map generated here indicates much lower degree day sums for various parts of the Antarctic Peninsula than that modelled by Chown et al.[12], highlighting the potential mismatch between satellite-derived climate variables and field measurements made at a smaller spatial scale. However, they both concur in highlighting that the north-western part of the Antarctic Peninsula is warmest and at the highest risk of the establishment by non-native species, as supported by reported occurrences from this region[33]. Applying the warming scenarios to all successfully overwintering species considered in the study led to post-germination growth periods of 28–132 and 73–149 days (at ~+3 and +5 °C warming, respectively) (Fig. 3). Future climate warming along the Antarctic Peninsula is also unlikely to be uniform due to the mountainous topography where local (km scale) conditions will unlikely be accurately represented in atmospheric models that are typically currently available at 100 km resolution[56], as was also documented during recent warming periods in this region[4]. Our uniform warming approach does, however, provide an indication of how changes in soil thermal conditions can affect germination and growth of non-native plants and thereby highlights which plant types/taxa are most likely to establish.

We recognise a number of caveats in the assumed climate suitability of the Antarctic Peninsula region for the tested plant types, as microclimates cannot be reproduced in detail in the climate chambers used, a limitation that applies to all such studies[57,58]. Further improvements in this type of methodology

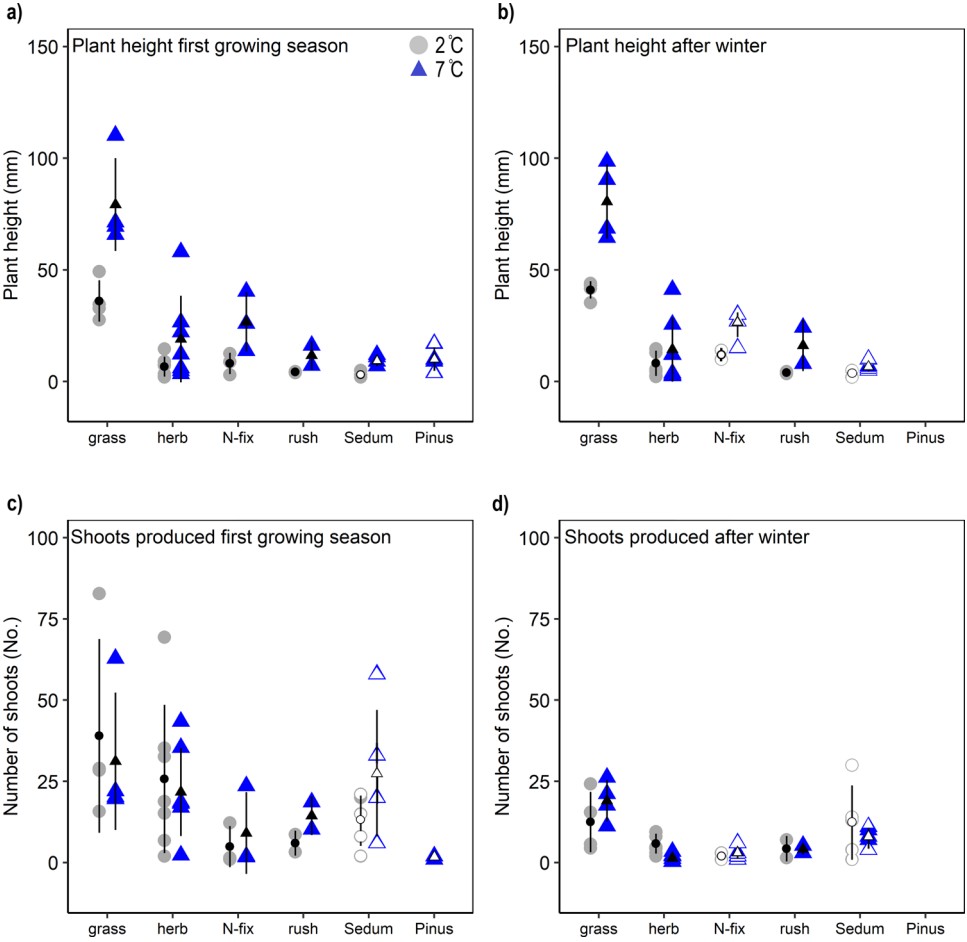

**Fig. 2 Plant growth responses to temperature.** Plant height and the number of shoots of the main plant types under Antarctic growing season temperatures (2 °C; grey circles) and climate warming (7 °C; blue triangles) in Antarctic soil. **a** Plant height at the end of the first growing season. **b** Plant height at the end of the second growing season after the simulated winter period. **c** Number of plant shoots produced at the end of the first growing season. **d** Number of plant shoots produced at the end of the second growing season after the simulated winter period. Grey circles and blue triangles represent species-specific means within each plant type (grass: 4, herb: 7, N-fix: 3 and rushes: 2), with the exception of *Sedum album* and *Pinus sylvestris* where 'open' data points represent experimental pots (n = 4-5) and for N-fixing plants after winter when only *Astragalus cruckshanksii* grew (n = 4-5). Black symbols represent the mean with SD as error bars. For a complete species list see Supplementary Tables 1 and 2.

are clearly possible when technological limitations can be overcome. Climate chambers such as those used in our study cannot mimic summer freezing events and can achieve only part of the diurnal and seasonal temperature variability that exists in Antarctica[26], factors that could affect species survival[59]. For instance, the mean soil surface winter temperatures at Mars Oasis and Coal Nunatak are around −15 °C (Supplementary Table 5) indicating that snow accumulation and insulation at the measuring station is minimal, as further highlighted by the occurrence of minimum winter surface temperatures of −38 °C[26]. Further factors, such as high wind and associated transpiration losses, that may affect the survival of any of the tested species, also cannot be reproduced in chamber experiments[60]. Light regimes will not change as climate warming increases temperature, although could be influenced by changes in cloudiness (in turn likely related to precipitation). There may, therefore, be a mismatch for plant growth between optimal temperature and light conditions as invasive species expand their range southwards[61]. However, given that the simulated light regimes used in this study were based on field measurements of PAR at 67°S it is likely that light conditions north of this will still be suitable, with the study findings, therefore, applying at minimum

for the larger part of the Antarctic Peninsula. Accepting such caveats, the data presented in this study show that the potential for seeds of non-native species to germinate and survive through the annual cycle along the Antarctic Peninsula under current and predicted future climate scenarios is much greater than previously realised. The survival of species at specific Antarctic locations will then largely depend on small-scale topographic variations, such as created by boulders and north-facing crevices, that may provide sufficient protection against wind and freezing events at even the most southern locations[60,62].

Our study shows experimentally that, both in terms of the number of species and of geographical range, there is greater potential for species to establish on the Antarctic Peninsula than indicated to date by modelling approaches that have only been applied to a small number of species[16]. Temperature records available from the southern locations of Mars Oasis and Coal Nunatak (~72°S) indicate that contemporary soil surface degree day sums would be sufficient at those sites for various non-native species to germinate, again further south than distribution modelling approaches have yet indicated[16,29]. The same pattern is also reflected a greater extent under the simulated warming scenario tested. Given the divergent methodologies and

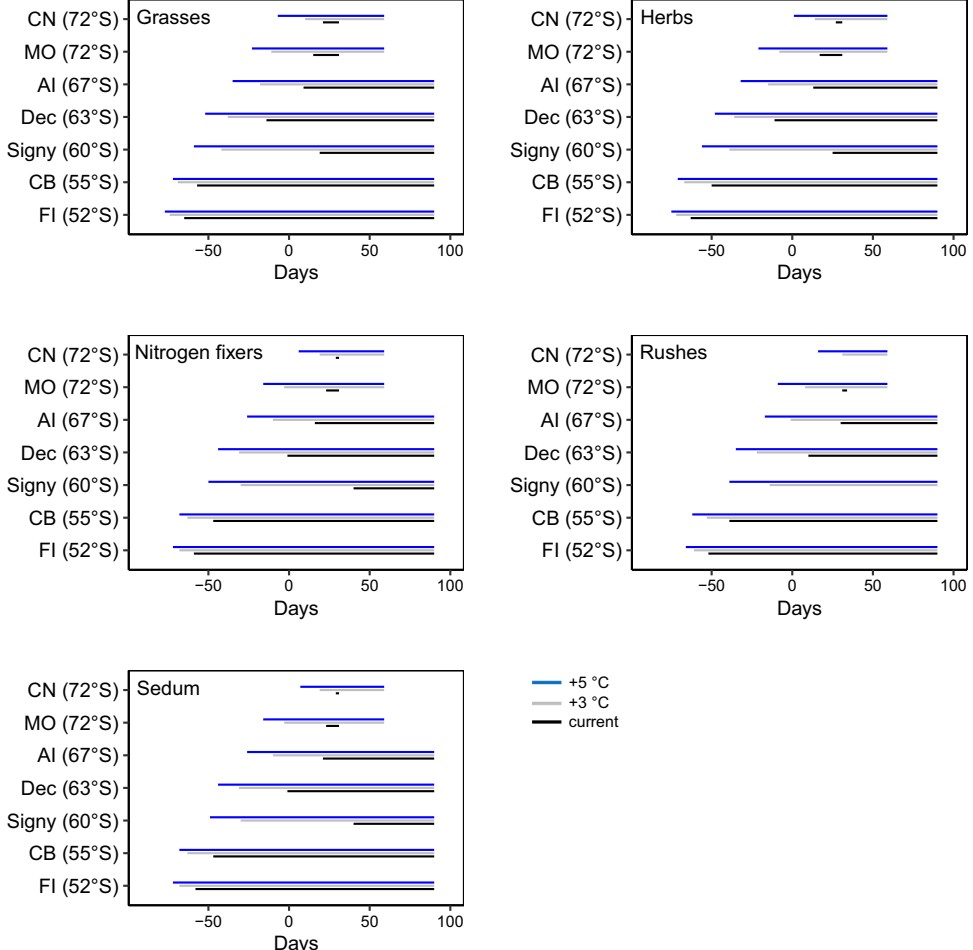

**Fig. 3 Potential time available for plant germination and growth in the Antarctic Peninsula.** Calculated date of germination and growth duration for different plant types under local soil surface climate conditions and climate warming scenarios at locations from the Falkland Islands (52°S) to Coal Nunatak (72°S). 0-Days = 1st January. Growing season ends on 31st March due to diminishing light conditions at the Falkland Islands (FI), Cerro Bandera (CB), Signy, Deception (Dec) and Anchorage Island (AI), while freezing commences at the end of February at Mars Oasis (MO) and Coal Nunatak (CN) under the climate warming scenario.

underlying assumptions between seed viability testing and modelling approaches such different outcomes are not surprising but do highlight the need to consider both approaches in assessing the risk of non-native species establishment and biological invasions.

These findings also highlight the potential danger from the large number of non-native plant species that are already established on sub-Antarctic islands[23,44,63], in particular South Georgia which lies on the Scotia Arc, that may act as stepping stones for species to reach the Antarctic Peninsula. Our data also indicate that typical Antarctic Peninsula soil[64,65], as used in this study, provides a suitable substrate to support non-native plant growth if seeds can reach these locations. The probability of species reaching suitable establishment sites along the Antarctic Peninsula to a very large extent depends on the effectiveness of mitigation measures adopted to minimise the risk of human-assisted introductions[7,12]. However, natural dispersal and the overall lowering of environmental barriers against non-native species dispersal to and establishment in Antarctica[16,66,67] mean that viable propagules of various regionally non-native plants and other species, particularly from southern South America, will still be likely to reach suitable sites for germination. Thus, ongoing and robust monitoring is required for the early identification and removal of such species.

## Methods
To test whether seeds of non-native plant species could germinate and grow in Antarctic soil under current conditions we conducted a climate chamber experiment where germination time and subsequent growth of 26 species was quantified (Supplementary Table 1). The experiment ran for two growing seasons with an intervening six month simulated winter period (−5 °C in darkness). The climate chamber was set to 2 °C, approximating mean growing season soil surface temperatures measured at both Signy Island (60°S) and Anchorage Island (67°S)[32]. Diurnal variation in soil temperature and light conditions were adapted every month to mimic the seasonal variation in field microclimate conditions (Supplementary Fig. 1)[32,68], see details below. To quantify the impacts of climate warming on the time required for germination and growth, a parallel climate chamber was run at 7 °C. The 5 °C temperature increase reflects the Representative Concentration Pathway 8.5 global climate warming scenario[69], which is appropriate given that the northern Antarctic Peninsula region already warmed by ~3 °C in the second half of the 20th Century and warming trends of ~0.5 °C/decade have been reported and are predicted for the remainder of the 21st Century[4,70].

We used soil obtained from beneath moss vegetation on Anchorage Island (Ryder Bay, south-east Adelaide Island, 67°34′S 68° 07′W) collected during January 2018 and transported frozen (−20 °C) to laboratories in the Netherlands. The soil was thoroughly mixed after thawing and divided across 260 pots (5 cm diameter and 5 cm height) to a depth of 3 cm (average 'soil' depth on Anchorage island). Seeds were added from 26 different plant species (see below for details on species selection and Supplementary Table 1 for species list, plant types used and seeding density) to 10 replicate pots per species. Seeding density ranged between 3 and 200 seeds/experimental pot due to differences in total number of seeds available and seed size.

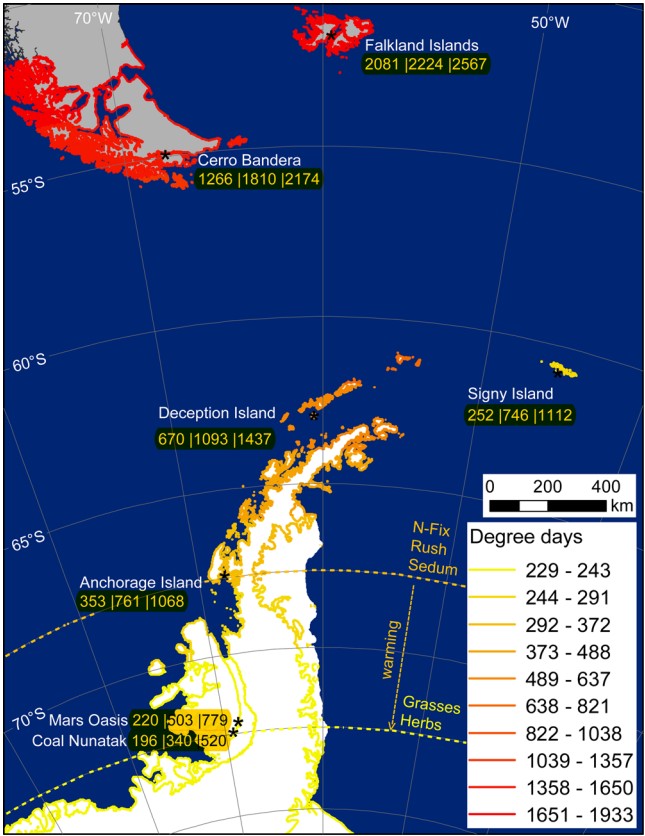

**Fig. 4 Current soil surface degree day sums for coastal ice-free regions along the Antarctic Peninsula.** Location names indicate where soil surface temperature was recorded (see Supplementary Table 5 and Fig. 3). Values represent: measured degree day sums and recalculated values following warming scenarios of +3 or +5 °C (current|+3 °C|+5 °C). The curved dashed line indicates the potential southern germination limit for the tested non-native plant types, based on the experimental study, with the arrow indicating the likely shift in germination limit as temperatures rise. No extrapolations on soil degree day sums were made beyond Coal Nunatak due to lack of temperature records further south. Therefore, species limits are bound by the same restrictions. Signy Island has a much lower degree day sum than would be expected from its latitude due to high cloud cover and this may also be the case at other extrapolated locations. Note that much of the Antarctic Peninsula coastline is currently not free from ice and the degree day sums reflect potential values if no ice was present.

All pots were placed in a dark climate chamber (2 °C) for 6 weeks to allow for cold stratification. The seeds were moistened using a plant sprayer once each week. After stratification, the pots were divided between two climate chambers at the two (constant) temperatures of 2 and 7 °C. Each chamber, therefore, had five replicate pots per species. Plants were watered twice a week, using tap water, to ensure that soils remained moist. Total water given over the growing season was c. 150 mm, which is well within the range of measured precipitation values for sites along the Antarctic Peninsula[71,72]. To avoid the effects of placement within each chamber, pot positions are randomly repositioned every week and all pots were moved between chambers and chamber temperature adapted every month to avoid any systematic 'chamber effect' throughout the experiment.

As a positive control to our experiment, we assessed seed germination and growth under non-nutrient limited growing conditions by duplicating the experimental procedure described above using commercial potting soil (Horticoop Bleiswijk, The Netherlands; Potting soil No. 4). Finally, to test for any temperature limitations on seed germination and growth, we placed five replicate pots/species with potting soil and the same seeding density as used for the main experiment at 15 °C. *Eucalyptus coccifera* and *E. perriniana* germinated at 15 °C but failed to germinate at both 2 and 7 °C in either soil type (Supplementary Tables 6–8). Plants that germinated at 2 and 7 °C did so in both potting soil and Antarctic soil.

**Antarctic climate simulation.** The experiment ran for two simulated Antarctic growing seasons with 6 months of simulated winter conditions (−5 °C in darkness)

in between. The winter period is relevant as the establishment of seedlings requires them to survive Antarctic winter conditions. The results from this study are therefore most relevant for sites along the Antarctic Peninsula with sufficient snow accumulation, as frozen soil can experience temperatures much lower than −5 °C without snow insulation[26,73], which would kill many plant species[59]. Terrestrial microclimates in Antarctica are characterised by large diurnal and seasonal variation[26,53,68], which is challenging to simulate within climate chambers. To mimic natural field conditions, we used detailed micro-meteorological data recorded at Anchorage Island during 2004–2006[68] to provide baseline Antarctic climate conditions for the experiment. The mean summer (December–February) temperature recorded within moss vegetation is 2 °C, although it can reach up to 30 °C over short periods due to direct solar insolation. Solar radiation at Anchorage Island, quantified in the field with photosynthetically active radiation sensors (PAR, SKP215 Campbell Scientific UK), varies greatly across seasons and over the diurnal cycle, with zero light during mid-winter and summer peaks values of PAR above 2000 µmol/m²/s (Supplementary Fig. 2). To simulate these conditions we used commercial walk-in cooling chambers (THEBO Horeca) with RIVA Cold refrigeration units (Rivacold srl—Vallefoglia, Italy). Growing season air temperatures were set to 2 °C and diurnal light intensity was modulated through light-emitting diode (LED) lamps (Hortilight Sunfactor 270; 405 W) horizontally placed at 50 cm above pot height. The monthly mean diurnal light conditions recorded on Anchorage Island[68] were used for 4 weeks of spring (2 weeks at October and November light conditions), 3 summer months (December, January, February light conditions) and 4 autumn weeks (2 weeks at March and April light conditions) (see Supplementary Figs. 1 and 2 for light and temperature conditions). During the last two weeks of the simulated growing season, the temperature in both chambers was lowered to 1 °C. The mid-day light conditions considerably raised the soil surface temperature in the pots to levels comparable with those recorded in the field (Supplementary Fig. 1), while during darkness the soil/air conditions were equal to the chamber settings (2 or 7 °C). The maximum light levels reached within the climate chamber, during a Southern Hemisphere December mean diurnal cycle, were approximately 80% of those recorded in the field (1321 µmol/m²/s). Relative humidity was kept between 60 and 90%, similar to field conditions (Supplementary Fig. 1), by placing a water bath in the climate chamber. Relative humidity and air temperature were recorded at hourly intervals (HOBO U23 Pro v2, Bourne, MA, USA, in both climate chambers). Soil temperature was measured using button loggers (I-button, Maxim Integrated, San Jose, CA, USA) at hourly intervals in eight experimental pots, two with Antarctic soils and two with potting soil in both 2 and 7 °C chambers, not containing any seeds.

After the simulated autumn, all plants were placed in a dark freezing chamber at −5 °C for 6 months, which simulates Antarctic sub-nivean winter conditions where sufficient snow accumulates to insulate the soil from freezing ambient temperatures[26,32]. Pots were sealed within plastic bags to limit freeze-desiccation during the winter period. We note that high light in combination with freezing conditions, which can occur during spring and summer[68], was beyond the capabilities of the climate chambers. In addition, the refrigeration units went through a standard defrost cycle every 6 h, which raised the air temperature to approximately 5.0 °C (2 °C chamber) and 9.5 °C (7.0 °C chamber) for about 30 min, although this temperature anomaly was reduced during night-time and shoulder seasons when light intensity was lower.

**Plant species selection.** Species selections were based on either the proximity of native distributions to Antarctica (i.e. southern South America, southern Australia and New Zealand), or on possession of ruderal characteristics and boreal/Arctic/Alpine provenance, as human transport is known to be associated with the introduction of propagules from such regions[7,14,23]. Only species whose seeds were readily available through commercial channels or collections available to us were selected. We chose plant types with different growth strategies, to determine if specific types may have more potential in establishing in Antarctic terrestrial ecosystems. Overall, we selected 26 species across 18 families including trees (4), shrubs (4), grasses (4), rushes (2), fern (1), succulent (1) and herbs (10), with four of the latter being nitrogen-fixing plants, which are known for their high invasion potential[74] (Supplementary Table 1).

**Biotic measurements.** From the onset of spring, when lights came on, we noted the number of days required for the first seed to germinate in each experimental pot at 3–4 day intervals until the end of the first growing season. From this, we calculated the number of days required for each species to germinate as well as the degree day sums above 0 °C in the soil. Plants were left untouched irrespective of the number of emerged seedlings in each pot. The differences in seeding density between species and lack of thinning could in theory affect seedling emergence. However, there were no clear patterns of degree day sums for emergence with seeding densities within our data set. Therefore, we assume that the observed patterns of seedling emergence are in response to temperature treatments and not the result of the number of seeds or plant density. We counted the total number of shoots/leaves and maximum plant height within each experimental pot at the end of the first growing season. Leaves were counted for: grasses, herbs, rushes and N-fixing plants while for shrubs, trees and the succulent we counted shoots. Following 'winter' we quantified any further germination and growth as described in the first growing season. However, due to logistical constraints (lockdown) resulting from

the Covid-19 outbreak, we were forced to stop the experiment after 100 days (instead of the intended 140 days) of the second simulated growing season, and the final measurements of the number of shoots/leaves and plant height had to be made before the end of the growing season (January–February).

**Degree day sums for plant germination along the Antarctic Peninsula.** To estimate the likelihood that the selected non-native species would be able to grow in the maritime Antarctic we quantified soil surface degree day sum (>0 °C) accumulation during the growing season at Signy Island (60.72°S), Deception Island (62.58°S), Anchorage Island (67.36°S), Mars Oasis (71.98°S) and Coal Nunatak (72.05°S) from soil surface temperature records[32,75–77]. Soil surface temperature data from the Falkland Islands (51.76°S) and Patagonia (Cerro Bandera on Isla Navarino, 800 m asl. 54.56°S) in southern Chile were included to provide a reference comparison with regional cool temperate sites at which some of the selected species naturally occur[30,78]. The degree-day sum requirements for seed germination of each plant species were obtained from the laboratory study (of the first growing season at 2 °C), and the remaining growth period until freezing was calculated for each of the locations from the soil temperature data. This approach was also used for the +5 °C warming scenario, where hourly temperature records were increased by 5 °C, assuming that air warming will be reflected in the soil, and degree-day sum accumulation was recalculated for each location. This uniform warming approach, although directly linked to the experimental work, may not reflect variations in warming intensity as anticipated across the length of the Antarctic Peninsula, similar to already documented variation in warming trends in this region[4,79]. The use of a bioclimatic envelope is however appropriate for the spatial scale under consideration[80], especially given the limited availability of soil surface temperature records along the Antarctic Peninsula. The end of the growing season was defined as 31st March due to diminishing light conditions at the Falkland Islands, Patagonia, Signy, Deception and Anchorage Islands, while soil freezing started at the end of January at Mars Oasis and Coal Nunatak (and at the end of February under the +5 °C warming scenario). There was a strong correlation ($r^2 = 0.971$) between measured soil surface degree day sums (>0 °C) and latitude across sites along the Antarctic Peninsula (Supplementary Fig. 3). We used this correlation to map degree day sums along the Antarctic Peninsula and calculate how this may change under future warming scenarios of +3 and +5 °C. Although this correlation is based on a limited number of measuring stations and there are likely to be site-specific deviations, the degree-day patterns are in line with declining air temperature with increasing latitude along the Antarctic Peninsula[52]. Note that Signy Island (60.7°S) did not fit the overall pattern due to the typically high cloud cover at this location[53], resulting in much lower degree day sums for its latitude, and was therefore omitted from this correlation.

**Statistics and reproducibility.** Factorial ANOVAs were used to make species-specific and plant type comparisons of time required for germination, numbers of shoots/leaves and plant maximum height between 2 (current) and 7 °C (warmed) experimental treatments. Plant species were used as replicate units in the statistical analyses unless only one species was available with a certain plant type and experimental pots ($n = 5$) were used as replicate units. Homogeneity of variance was visually inspected through plotting residuals versus fitted values and log-transformations were applied when required. Statistical analyses were performed using R[81].

**Reporting summary.** Further information on research design is available in the Nature Research Reporting Summary linked to this article.

## Data availability

The authors declare that the plant germination and growth data supporting the findings of this study are available within the paper and its supplementary information files. Location-specific degree day sums are available from the Netherlands Polar Data Center (Dataset—Plant germination in Antarctica|Netherlands Polar Data Center (npdc.nl)[82]. Source data for Figs. 1–3 are provided with the paper.

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

## Acknowledgements

We are grateful for the logistical support given by the British Antarctic Survey during the fieldwork. We would like to thank Richard van Logtestijn and Rob Broekman for their help during the laboratory experiment. We are grateful to Laura Gerrish (BAS Mapping and Geographic Information Centre) for her assistance in the preparation of Fig. 4. This work was funded by a grant from the Netherlands Polar Programme (NPP-NWO 866.16.006) and by Natural Environment Research Council core funding to the British Antarctic Survey "Biodiversity, Evolution, and Adaptation" team. Angélica Casanova-Katny appreciates the logistical support of the Instituto Antártico Chileno (INACH, project FR-04-18) and the staff of the Spanish Gabriel de Castilla scientific base during the fieldwork on Deception Island, financed by the project ANID-FONDECYT 1181745.

## Author contributions

S.B., P.C. and R.A. conceived the study and S.B. carried out the experiment. S.B., P.C., A. C.-K. and R.A. contributed to the writing of the paper.

## Competing interests

The authors declare no competing interests.
