## [Peer Review File · Communications Biology]

This manuscript has been previously reviewed at another Nature Research journal. This document only contains reviewer comments and rebuttal letters for versions considered at Communications Biology.

REVIEWERS' COMMENTS:

Reviewer #1 (Remarks to the Author):

Dear Authors

I have reviewed your revised manuscript and believe you have done a very thorough job addressing my previous concerns (thank you).

I think you have addressed my major concern about extrapolating soil temperature data well and have included adequate text highlighting the limitations of the degree day mapping.

The manuscript also reads much more intuitively with the restructure.

I have only two further minor comments:

Fig. 4. - It would be good to explicitly mention that the map relates to current degree day sums (it is implied, but might confuse some readers).

Amending to "Current soil surface degree day sums for coastal..." would work.

Supp Fig. 3. - reword 'model' to 'calculate' the same as you did in the main text

Reviewer #2 (Remarks to the Author):

This paper is an important contribution to the literature on the invasion risk to the Antarctic, where climate change is significantly increasing the risk for invasion. The importance of this paper lies in its experimental component which has been little explored for the region; mostly, invasion risk has only been quantified from a climate matching/SDM approach. The authors use a rigorous experimental design and the results of this paper should be interesting to a wide audience working in the Polar and other cold regions.

The paper has been well revised since the previous round of revisions. I think it is mostly in a good condition, though I have some comments. (I was not a reviewer on a previous version of this paper.) I think that some aspects need clarification. (I hope the authors excuse my pedantry on some grammatical issues; I do think the (fairly few) suggestions I make below will improve the text.)

The one aspect of the ms by which I would not be fully convinced is the attempt at generalising how different functional groupings (e.g. grasses vs rushes vs trees, etc) show different risks of invasion to the region of interest. Species across the different groups to be used in the experiment were not consistently/rigorously selected, so it is difficult to know whether the patterns found really hold for a group, or just hold for the few spp that happened to be selected for this project. Also, while the authors find plant functional type is a significant predictor of the response variables they measured, they do not show post-hoc results of their ANOVAs, which would indicate which functional groups differ from one another. I am not suggesting that authors remove this section, but I would recommend that they strongly indicate the caveats around the assumptions they are making for the analyses, and also that they present the post-hoc results and include the findings from these in their analyses of the data.

Minor comments:

I would have liked to have seen some background on the criteria that were used to select the plants before the methods section. Below are the comments I made while reading the paper from start to finish, to illustrate my confusion about what spp were selected to run this experiment until I read the methods section right at the end of the ms.

- Line 48: I would recommend providing some information on the plants selected, e.g. are they common invasives to the Antarctic region, or widespread global invaders, or particularly aggressive invaders?
- Line 60: related to the above comment, it is difficult to assess, based on 26 species, which global invaders may become major invaders to the sub-Antarctic. This sentence suggests that major generalisations on which groups will become invasive can be made from such a small sample size; I would argue not. Therefore, I would recommend keeping it more general here. If my first comment above is addressed, then this could be easier. E.g. if spp selection was done based on current range of spp (widespread invaders were selected), then this could be rewritten as "which widespread global invaders are most likely to establish, given opportunity..."
- Line 334: OK, I see that there are some details on spp selection here. I would highly recommend that the basics of this are given in the introduction to provide some context to the spp that were selected. Also, several of the spp selected here are invaders on sub-Antarctic islands. Would this be an additional reason to list for the selection of these species?

Line 77-79: unclear whether this is referring to the experiment in Antarctic soils or in potting soil.

Line 80: I find this newly inserted section abrupt; it doesn't flow well with the rest of the text. Also, it fails to clearly indicate that the results that there was no germination could mean one of two things (see below). Perhaps the section could be rewritten as something like this, where the two possible reasons for the findings are more clearly differentiated:

"None of the germinating species set seed during the course of the experiment under either scenario. This could suggest that these species are unable to set seed under Antarctic climatic conditions. Such a scenario would not, however, prevent species persisting and expanding their populations as clonal growth is often common in colder habitats³² and has been observed for both native Antarctic vascular plants³³⁻³⁵, the spread of the non-native grass *P. annua* at various sites in Antarctica^{36, 37}, and its congener *P. pratensis* at its single (now eradicated) occurrence site on the Antarctic Peninsula. Alternatively, our finding could be a result of plants not having reached a sufficient size to set seed. Indeed, both initial germination and survival and development to larger plant size is a crucial factor for establishment of non-native vascular plants in Antarctica. This is evident from the fact that few seed germination experiments that were conducted in Antarctica (before the current general ban on the import of non-native species under the Protocol on Environmental Protection to the Antarctic Treaty³⁸) did not show success, while transplants typically fared better³⁹."

Line 84: is there a reference for this statement about *P. pratensis*? (If not, it's probably not essential.)

Line 93-94. "This suggests..." Again, a slight grammatical issue: at first reading "This" seems to refer to the findings for both grasses and *C. arvensis* presented in the preceding sentence, whereas in reality it only refers to the finding for *C. arvensis*. I would suggest splitting the preceding sentence (lines 91-93). The first sentence can be about grasses, the second about *C. arvensis*, and then the sentence in lines 93-94 could be made part of this second sentence so that all info on *C. arvensis* is in one sentence.

Table 1: Unclear what "T x Pt" represents? Also, "PT" is abbreviated in the table heading, but does not occur in the table. Additionally, I assume the numbers in the table headings represent DFs? Perhaps this could be specified? Finally, in the heading it is stated that the number of leaves is shown, but this is not seen in the table.

Also, in Table 1 it is shown that Plant type is a significant predictor of the response variables tested; however, no posthoc results seem to be shown, so that it is not clear which plant functional types differ from which other types? See my comment above.

Line 100: I would avoid using "The same pattern" here, as two different response variables are being compared here. Instead, consider a phrase such as "Similarly..."

Line 102: "With limited differences between other plant species" Unclear: differences in what?

Line 102-104: Can you refer the reader to a figure and/or statistical table where these differences were tested.

Line 105: I would recommend writing out what is meant by "both scenarios" here. Both current and

+5C scenarios?

Line 113: What is meant by "time difference"?

Line 125: What is meant by "slow" germination here? Many trees can germinate very fast (in my experience, some within 2 days of planting the seed.) I would be careful of stating that a slow rate of germination is a limiting factor of trees in the region.

Line 125-126: I would suggest generalising this statement to "This reflects in part, the temperature constraints on trees in cold regions⁴³". This experiment was conducted on the level of the individual plant, tree lines represent landscape characteristics.

Line 284-286: grammatically, this sentence is clumsy, as the temperatures are not subject of the first part of the sentence, but are assumed as subjects of the second part of the sentence. I would recommend breaking this up into 2 sentences.

What happened when more than 1 plant germinated in a pot? Were all other plants removed from the pots?

The number of seeds planted per pot differed between species. If there were more seeds planted in one pot, could this result in a faster time to germination as a result of the fact that, by chance, there were more viable seeds, and thus a higher chance that one of them would germinate earlier?

Similarly, if plants were not thinned in pots where more than one seed germinated, did all plants in the pots re-emerge earlier in the next season, or was the earlier re-emergence due to there being more plants, resulting in greater variability in re-emergence times, with earlier re-emergence in those pots with many plants?

Line 303: Supplementary Figure S1 does not display PAR values. Should this be "Figure S2"?

Lines 348-349. It states that either leaves or shoots were counted? When was a shoot counted and when a leaf? How was a shoot defined?

Figure 1 caption:

- Line 608: "day", not "say" sums.
- Line 608: How is b different from d? From the figure caption it sounds like they are identical.
- I would recommend using different colours and/or symbols where the symbols mean something different. It is confusing if sometimes a blue triangle represents a spp mean and sometimes an individual pot. The same with the error bars. Do the error bars of Pinus and Sedum represent means across pots for only one spp, while for grasses, herbs, etc they represent means across pots of all different spp? Of do the latter represent means across spp rather than pots? (This comment is also relevant to other figures)

Figure 4

- caption – the first and second sentences seem to be repeats of the same sentence
- it is indicated that values represent +3 or +5C temp increases – unclear how to differentiate these on the figure
- "Curved lines..." The curved lines are not distinguishable from the lines representing degrees latitude. Perhaps they could be redrawn in a different colour?
- I struggle to understand the arrow extending from rushes, Sedum and N-fixers to grasses and herbs. The caption seems to indicate that this arrow shows the shift in germination limit with temp increases. However, this arrow is only shown for the rushes etc, and not the grasses etc. Also, it seems unlikely that the germination limit for rushes etc under a warming scenario will be at the exact position of that of the grasses under current conditions.
- It is difficult to read the black text next to Mars Oasis and Coal Nunatak, and unclear what this text is referring to.

Supplementary Figures S1 and S2: what does the x-axis represent?

Supplementary Tables 3 and 4: from the figure heading it appears that means were either calculated based on only 5 pots, or that there were no pots. In contrast, in Table S2, means were calculated based on 1-5 pots, or there were no pots. Why is this different between Tables S3/4 and S2?

Response to reviewers Communications Biology

Dear editor,

thank you kindly for allowing us to revise our manuscript titled "*Warming impacts potential germination of non-native plants on the Antarctic Peninsula*". We have modified our manuscript following the suggestions and comments raised by the reviewers and the editorial aspects as can be found in our detailed response below.

Looking forward to your response.

Kind regards,
Stef Bokhorst

REVIEWERS' COMMENTS:

Reviewer #1 (Remarks to the Author):

Dear Authors

I have reviewed your revised manuscript and believe you have done a very thorough job addressing my previous concerns (thank you).

I think you have addressed my major concern about extrapolating soil temperature data well and have included adequate text highlighting the limitations of the degree day mapping. The manuscript also reads much more intuitively with the restructure.

I have only two further minor comments:

Fig. 4. - It would be good to explicitly mention that the map relates to current degree day sums (it is implied, but might confuse some readers).

Amending to "Current soil surface degree day sums for coastal..." would work.

Reply: we have added "current" to the figure heading as suggested

Supp Fig. 3. - reword 'model' to 'calculate' the same as you did in the main text

Reply: we have used "calculate" to the figure heading as suggested

Reviewer #2 (Remarks to the Author):

This paper is an important contribution to the literature on the invasion risk to the Antarctic, where climate change is significantly increasing the risk for invasion. The importance of this paper lies in its experimental component which has been little explored for the region; mostly, invasion risk has only been quantified from a climate matching/SDM approach. The authors use a rigorous experimental design and the results of this paper should be interesting to a wide audience working in the Polar and other cold regions.

The paper has been well revised since the previous round of revisions. I think it is mostly in a

good condition, though I have some comments. (I was not a reviewer on a previous version of this paper.) I think that some aspects need clarification. (I hope the authors excuse my pedantry on some grammatical issues; I do think the (fairly few) suggestions I make below will improve the text.)

The one aspect of the ms by which I would not be fully convinced is the attempt at generalising how different functional groupings (e.g. grasses vs rushes vs trees, etc) show different risks of invasion to the region of interest. Species across the different groups to be used in the experiment were not consistently/rigorously selected, so it is difficult to know whether the patterns found really hold for a group, or just hold for the few spp that happened to be selected for this project. Also, while the authors find plant functional type is a significant predictor of the response variables they measured, they do not show post-hoc results of their ANOVAs, which would indicate which functional groups differ from one another. I am not suggesting that authors remove this section, but I would recommend that they strongly indicate the caveats around the assumptions they are making for the analyses, and also that they present the post-hoc results and include the findings from these in their analyses of the data.

Reply: Thank you for the positive comments on our manuscript. We have clarified, throughout the ms where there are clear differences in plant types with regard to degree day sum requirements and no. days till emergence (see lines 107-110 and 123-125). We did not mention this in the previous manuscript, because nearly all of the significant differences were due to the much slower emergence of Pinus compared to the other plants.

Minor comments:

I would have liked to have seen some background on the criteria that were used to select the plants before the methods section. Below are the comments I made while reading the paper from start to finish, to illustrate my confusion about what spp were selected to run this experiment until I read the methods section right at the end of the ms.

- Line 48: I would recommend providing some information on the plants selected, e.g. are they common invasives to the Antarctic region, or widespread global invaders, or particularly aggressive invaders?

Reply: we have added the following lines to the introduction: “For this work we selected species based on 1) a native distribution in the proximity of Antarctica, 2) ruderal characteristic, or 3) those that are invading sub-Antarctic islands” (see lines 49-51).

- Line 60: related to the above comment, it is difficult to assess, based on 26 species, which global invaders may become major invaders to the sub-Antarctic. This sentence suggests that major generalisations on which groups will become invasive can be made from such a small sample size; I would argue not. Therefore, I would recommend keeping it more general here. If my first comment above is addressed, then this could be easier. E.g. if spp selection was done based on current range of spp (widespread invaders were selected), then this could be rewritten as “which widespread global invaders are most likely to establish, given opportunity...”

Reply: We have rewritten this sentence as suggested (see lines 62-63)

- Line 334: OK, I see that there are some details on spp selection here. I would highly recommend that the basics of this are given in the introduction to provide some context to the spp that were selected. Also, several of the spp selected here are invaders on sub-Antarctic islands. Would this be an additional reason to list for the selection of these species?

Reply: We have added additional information on this in the introduction as per the earlier comments (see lines 49-51).

Line 77-79: unclear whether this is referring to the experiment in Antarctic soils or in potting soil.

Reply: this applied to both potting soil and Antarctic soil for those three species. The sentence now reads: “However, *Betula nana*, *Blechnum penna-marina* and *Larix sibirica* failed to germinate in potting soil, suggesting that for these species the lack of germination in Antarctic soil may have been due to inherently poor germination” (See lines 80-81)

Line 80: I find this newly inserted section abrupt; it doesn't flow well with the rest of the text. Also, it fails to clearly indicate that the results that there was no germination could mean one of two things (see below). Perhaps the section could be rewritten as something like this, where the two possible reasons for the findings are more clearly differentiated:

“None of the germinating species set seed during the course of the experiment under either scenario. This could suggest that these species are unable to set seed under Antarctic climatic conditions. Such a scenario would not, however, prevent species persisting and expanding their populations as clonal growth is often common in colder habitats³² and has been observed for both native Antarctic vascular plants³³⁻³⁵, the spread of the non-native grass *P. annua* at various sites in Antarctica^{36, 37}, and its congener *P. pratensis* at its single (now eradicated) occurrence site on the Antarctic Peninsula. Alternatively, our finding could be a result of plants not having reached a sufficient size to set seed. Indeed, both initial germination and survival and development to larger plant size is a crucial factor for establishment of non-native vascular plants in Antarctica. This is evident from the fact that few seed germination experiments that were conducted in Antarctica (before the current general ban on the import of non-native species under the Protocol on Environmental Protection to the Antarctic Treaty³⁸) did not show success, while transplants typically fared better³⁹.”

Reply: We have incorporated the suggested changes to these lines (see lines 82-96).

Line 84: is there a reference for this statement about *P. pratensis*? (If not, it's probably not essential.)

Reply: Citation is now included on line 90 (Perterra L, Hughes K, Tejado P, Enríquez N, Lucíañez M, Benayas J (2016) Eradication of the non-native *Poa pratensis* colony at Cierva Point, Antarctica: A case study of international cooperation and practical management in an area under multi-party governance. *Environmental Science & Policy* 69:50-56 doi:10.1016/j.envsci.2016.12.009)

Line 93-94. “This suggests...” Again, a slight grammatical issue: at first reading “This” seems to refer to the findings for both grasses and *C. arvense* presented in the preceding sentence, whereas in reality it only refers to the finding for *C. arvense*. I would suggest splitting the preceding sentence (lines 91-93). The first sentence can be about grasses, the second about *C. arvense*, and then the sentence in lines 93-94 could be made part of this second sentence so that all info on *C. arvense* is in one sentence.

Reply: these lines have now been split up and read as follows: “Sixteen of the 18 species showed regrowth in the second summer season following exposure to simulated winter conditions with grasses growing from overwintering roots. *Cerastium arvense* had above-ground plant parts that survived the winter which makes it a potentially high risk invader for Antarctic terrestrial ecosystems if seeds reach the Antarctic Peninsula.” (see lines 97-103)

Table 1: Unclear what “T x Pt” represents? Also, “PT” is abbreviated in the table heading, but does not occur in the table. Additionally, I assume the numbers in the table headings represent DFs? Perhaps this could be specified? Finally, in the heading it is stated that the number of leaves is shown, but this is not seen in the table.

Also, in Table 1 it is shown that Plant type is a significant predictor of the response variables tested; however, no posthoc results seem to be shown, so that it is not clear which plant functional types differ from which other types? See my comment above.

Reply: PT is the abbreviation we use for ‘plant type’ see: “different plant types (PT: grass, herb, nitrogen fixing plants, rushes, succulent and *Pinus*)” (T) is now added to the table heading to indicate Temperature. From this we hope it is clear that T×PT represents the interaction between Temperature with Plant Type.

The numbers do indeed represent degrees of freedom and the table heading now includes: “Degrees of freedom are presented between brackets for each main factor and the interaction term.”

The total no. of shoots/leaves was quantified (depending on the plant species/type). This has now been clarified in the table, on lines 360-361 and in table heading of Table S4.

Significant differences between plant types existed only through the difference with *Pinus sylvestris* during the first growing season (now mentioned on lines 107-110). For the second growing season, where *P. sylvestris* was no longer present, there was a significant differences in dd-requirements between *Sedum* and the rushes (now mentioned on lines 123-125).

The significant Temperature×Plant type (T×PT) interaction was due to a difference (Tukey HSD $p < 0.05$) between rushes (at 2 °C) and grasses (at 7 °C). (this is now mentioned in the table heading.)

Any differences between plant types in height and no. shoots/leaves (table 1) are fully due to the differences in growth form of the study species (i.e., grass vs. succulent) and are not relevant within the context of the manuscript.

Line 100: I would avoid using “The same pattern” here, as two different response variables are being compared here. Instead, consider a phrase such as “Similarly...”

Reply: We disagree on this point. The sentence starting with “The same pattern” deals with the no. days for plants to emerge between warmed and Antarctic temperatures which is also what the previous sentence deals with, so this reflects the same response variable.

Line 102: “With limited differences between other plant species” Unclear: differences in what?

Reply: This sentence now reads “while there were no significant differences between the other plant species for duration and degree day sums.” (see lines 108-109)

Line 102-104: Can you refer the reader to a figure and/or statistical table where these differences were tested.

Reply: Yes, we now refer to (Table 1, Fig. 1) as these show the mean values for days and DD-sums for the plant types.

Line 105: I would recommend writing out what is meant by “both scenarios” here. Both current and +5C scenarios?

Reply: This has now been adopted (see lines 112-113)

Line 113: What is meant by “time difference”?

Reply: We have added “until plant emergence” to clarify (see line 121-122)

Line 125: What is meant by “slow” germination here? Many trees can germinate very fast (in my experience, some within 2 days of planting the seed.) I would be careful of stating that a slow rate of germination is a limiting factor of trees in the region.

Reply: That sentence now read “However, even with these constraints, our data show that certain plant types (trees and shrubs) are less likely to establish due to the long time for plant emergence under current Antarctic conditions”. (see lines 133-136) Although some trees may germinate very fast in their own habitat, the tested species we used certainly did not do so under current Antarctic conditions or even the warming scenario.

Line 125-126: I would suggest generalising this statement to “This reflects in part, the temperature constraints on trees in cold regions⁴³”. This experiment was conducted on the level of the individual plant, tree lines represent landscape characteristics.

Reply: This suggestion has been adapted in the manuscript (see lines 135-136).

Line 284-286: grammatically, this sentence is clumsy, as the temperatures are not subject of the first part of the sentence, but are assumed as subjects of the second part of the sentence. I would recommend breaking this up into 2 sentences.

Reply: These lines have been split up as suggested and now read: “*Eucalyptus coccifera* and *E. perriniana* germinated at 15 °C but failed to germinate at both 2 °C and 7 °C in either soil type. Plants that germinated at 2 °C and 7 °C did so in both potting soil and Antarctic soil.” See lines (293-295)

What happened when more than 1 plant germinated in a pot? Were all other plants removed from the pots?

Reply: No, all plants were left untouched. This is now mentioned on lines 357-358.

The number of seeds planted per pot differed between species. If there were more seeds planted in one pot, could this result in a faster time to germination as a result of the fact that, by chance, there were more viable seeds, and thus a higher chance that one of them would germinate earlier? Similarly, if plants were not thinned in pots where more than one seed germinated, did all plants in the pots re-emerge earlier in the next season, or was the earlier re-emergence due to there being more plants, resulting in greater variability in re-emergence times, with earlier re-emergence in those pots with many plants?

Reply: In theory all the above may apply. However, the data from our study do not support this, as timing till emergence did not differ between plant types, if we exclude *Pinus sylvestris*, which you would expect if seeding density would play a role. If we zoom in on specific plant types we see that the degree days sums required till emergence among grasses varies quite a lot (92-194) despite all of them starting with 100 seeds, indicating that the patterns are driven by species-specific differences and not seeding density. For the herbs we also see large differences in DD-sums but there is no consistent pattern with seeding densities (ranging between 30-100). When looking only at the ‘after winter growth period’ there is also no clear pattern in seeding density with any species responses to temperature (while there

were significant differences of these species to temperature treatment; Supplementary table 2). So, although we acknowledge that seeding density and lack of thinning may affect seedling emergence in general, there is no strong data to support that it played a role in our experiment. Therefore, we would argue not to elaborate on this subject in the manuscript.

Line 303: Supplementary Figure S1 does not display PAR values. Should this be “Figure S2”?

Reply: Yes that is correct that should have been Supplementary Figure 2.

Lines 348-349. It is states that either leaves or shoots were counted? When was a shoot counted and when a leaf? How was a shoot defined?

Reply: . Leaves were counted for: grass, herb, rushes and N-fix while for shrubs, trees and *Sedum album* we counted shoots (see lines 360-361 and Supplementary table 4).

Figure 1 caption:

• Line 608: “day”, not “say” sums.

Reply: now corrected (see line 622)

• Line 608: How is b different from d? From the figure caption it sounds like they are identical.

Reply: These lines now read as follows: “**a** Number of days required to first germination and **b** number of days required for growth after winter. **c** Degree day sums required to first germination and **d** number of degree day sums required for growth after winter”

• I would recommend using different colours and/or symbols where the symbols mean something different. It is confusing if sometimes a blue triangle represents a spp mean and sometimes an individual pot. The same with the error bars. Do the error bars of Pinus and Sedum represent means across pots for only one spp, while for grasses, herbs, etc they represent means across pots of all different spp? Of do the latter represent means across spp rather than pots? (This comment is also relevant to other figures)

Reply: We have now used open symbols to represent pot values while the closed symbols still represent species means.

Figure 4

• caption – the first and second sentences seem to be repeats of the same sentence

Reply: The figure title now reads “**Current soil surface degree day sums for coastal ice-free regions along the Antarctic Peninsula**” and any duplication has been removed.

• it is indicated that values represent +3 or +5C temp increases – unclear how to differentiate these on the figure

Reply: We have added “(current | +3 °C | +5 °C).” to the figure heading to emphasize how the degree day sums are presented on the figure for each location (see line 666).

• “Curved lines...” The curved lines are not distinguishable from the lines representing degrees latitude. Perhaps they could be redrawn in a different colour?

Reply: We have redrawn these using thicker yellow-dashed lines.

• I struggle to understand the arrow extending from rushes, Sedum and N-fixers to grasses and herbs. The caption seems to indicate that this arrow shows the shift in germination limit

with temp increases. However, this arrow is only shown for the rushes etc, and not the grasses etc. Also, it seems unlikely that the germination limit for rushes etc under a warming scenario will be at the exact position of that of the grasses under current conditions.

Reply: This is a fair point. Given the previous concerns raised, regarding the spatial mapping of degree day sums from the limited available data for this region, we did not want to extrapolate beyond the most southern measuring location (Coal Nunatak). Therefore, we cannot make any predictions about the distribution of grasses and herbs under the warming scenarios and likewise, rushes and others cannot be plotted beyond Coal Nunatak. This is now mentioned in the figure heading “No extrapolations on soil degree day sums were made beyond Coal Nunatak due to lack of temperature records further south. Therefore, species limits are bound by the same restrictions” (see lines 669-671).

• It is difficult to read the black text next to Mars Oasis and Coal Nunatak, and unclear what this text is referring to.

Reply: We have placed the degree day sums (which was the text the reviewer is referring to) within a colored box.

Supplementary Figures S1 and S2: what does the x-axis represent?

Reply: this should have read ‘Hour’ as per the figure titles “**Season-specific diurnal microclimate patterns**” this has now been added to the x-axis.

Supplementary Tables 3 and 4: from the figure heading it appears that means were either calculated based on only 5 pots, or that there were no pots. In contrast, in Table S2, means were calculated based on 1-5 pots, or there were no pots. Why is this different between Tables S3/4 and S2?

Reply: This was a mistake; for all these table it should be: “Values are means of 1-5”